# Biosensors—Recent Advances and Future Challenges in Electrode Materials

**DOI:** 10.3390/s20123561

**Published:** 2020-06-23

**Authors:** Fernando Otero, Edmond Magner

**Affiliations:** Department of Chemical Sciences and Bernal Institute, University of Limerick, V94 T9PX Limerick, Ireland; fernando.otero.diez@ul.ie

**Keywords:** electrochemical biosensors, glucose biosensors, nanoporous metals, nanoporous gold, graphene, carbon nanotube, ordered mesoporous carbon, additive manufacturing

## Abstract

Electrochemical biosensors benefit from the simplicity, sensitivity, and rapid response of electroanalytical devices coupled with the selectivity of biorecognition molecules. The implementation of electrochemical biosensors in a clinical analysis can provide a sensitive and rapid response for the analysis of biomarkers, with the most successful being glucose sensors for diabetes patients. This review summarizes recent work on the use of structured materials such as nanoporous metals, graphene, carbon nanotubes, and ordered mesoporous carbon for biosensing applications. We also describe the use of additive manufacturing (AM) and review recent progress and challenges for the use of AM in biosensing applications.

## 1. Introduction

In comparison with other methods of detection such as optical, spectroscopic, and chromatographic, electrochemical sensors possess advantages such as simplicity, rapid response times, and high sensitivity [1]. Electrochemical sensors can be easily adapted for the detection of a wide range of analytes and can be incorporated into robust, portable, low cost, minituarized devices that can be tailored for particular applications [2]. Taking advantage of these attributes and the incorporation of highly specific biological recognition elements (enzymes, nucleic acids, cells, tissues, and so on), electrochemical biosensors are capable of selectively detecting a broad range of target analytes. As defined by IUPAC, electrochemical biosensors are “self-contained integrated devices, which are capable of providing specific quantitative or semi-quantitative analytical information using a biological recognition element (biochemical receptor), which is retained in direct spatial contact with an electrochemical transduction element” [3]. Bioelectrochemical sensors are used in environmental monitoring, healthcare, and biological analysis, among others. Depending on the recognition process, biosensors can be subdivided into two main categories: affinity and biocatalytic sensors. Affinity sensors operate via selective binding between the analyte and the biological component (i.e., antibody and nucleic acid) [4]. In contrast, biocatalytic devices incorporate enzymes, whole cells, or tissue slices that recognize the target analyte, and subsequently produce an electroactive species [5].

The first biosensor was described by Clark and Lyons in 1962 [6]. This biosensor was composed of an oxygen electrode, an inner oxygen semipermeable membrane, and a thin layer of glucose oxidase (GOx, EC 1.1.3.4) entrapped by a dialysis membrane. The decrease in the level of oxygen was proportional to the concentration of glucose resulting from the enzyme catalysed oxidation of β-D-glucose to β-D-glucono-δ-lactone [7]. Since this pioneering work, extensive efforts have been made to develop electrochemical biosensors for a wide range of analytes. Bioelectrochemical sensing devices have been effectively transferred from the laboratory to the point-of-care (POC) with global sales growing from less than $5 million per annum [8] thirty years ago to over $18 billion in 2018. Although commercial systems are available for a range of small molecules (lactose, uric acid, cholesterol, lactate, ketone, and so on), the market is dominated by glucose sensors, with approximately 90% of the market associated with glucose monitoring for diabetes [9]. Diabetes mellitus is one of the leading causes of death and disability in the world [10]. It is a metabolic disorder that causes insulin deficiency and hyperglycemia, resulting in blood glucose concentration deviating from the normal range of 3.9–6.2 mM [11]. According to the International Diabetes Federation (IDF), the number of diabetic patients increased from 151 million in 2000 to 415 in 2015. The IDF also predicted that the number of diabetic patients would increase to 642 million in 2040, with diabetes becoming the seventh-leading cause of mortality [12]. Commercial home use blood glucose sensors generally detect glucose in the concentration range of 1.1–33.3 mM with test times of less than 30 s [4]. GOx is widely employed as the recognition element in glucose biosensors owing its relatively low cost, high selectivity, and stability [13]. First and second generation sensors rely on the immobilization of the enzyme onto an electrode surface. As the redox active centers of GOx are at least 13–18 Å from the electrode surface, mediators are employed to shuttle electrons between the electrode surface and enzyme’s active site [14]. The direct oxidation of GOx occurs in third generation sensors, where the enzyme is specifically wired to minimize the distance between the active site of the enzyme and the electrode surface. Despite the considerable progress that has been made, the majority of commercial glucose sensors are based on second generation glucose sensors. The vast majority of commercial devices utilise blood samples obtained via a finger prick. The development of glucose biosensors based on the detection of glucose in fluids such as tears [15], saliva [16], and sweat [17] has been described. Such systems face challenges, in particular the poor correlation between glucose levels in blood and in other fluids and also significantly lower concentrations of glucose in fluids such as tears. Individually optimized designs must be developed [18] for commercially viable sensors, where challenges such as low cost, ease of manufacture, robustness, and portability are additional factors for consideration [19].

In contrast, detection of larger biomolecules such as nucleic acids and proteins faces significant additional challenges that include electrode fouling, non-specific adsorption of biological components at the electrode surface, lack of sensitivity in the appropriate concentration range, and in particular at low concentrations (femtomolar to attomolar) [8]. Commercial systems for the detection of larger biomolecules are dominated by pregnancy tests that rely on the detection of human chorionic gonadotrophin (hCG), a glycoprotein hormone secreted during pregnancy [20].

In this review, we describe recent advances on the use of materials as supports in electrochemical biosensors, and in particular the use of materials such as nanoporous metals, graphene, carbon nanotubes, and mesoporous carbon. Examples of the detection of clinically relevant molecules are provided, with a focus on the detection of glucose. An overview of invasive and non-invasive glucose monitoring with case studies is given. In addition, we discuss the use of additive manufacturing for electrochemical sensing applications.

## 2. Electrode Materials

Owing to their intrinsic conductivity, biocompatibility, and ease of manufacture, high surface area materials such as nanoporous gold, graphene, carbon nanotubes, and mesoporous carbon have been used for the preparation of electrodes for bioelectrochemical applications.

### 2.1. Nanoporous Metal Electrodes

Nanoporous metals are 3D bicontinuous structures with tuneable pore diameters and lengths that possess large surface areas, mechanical resistance, and high conductivity [21,22]. Although nanoporous electrodes have been prepared using a range of metals such as copper, silver, and palladium, the majority of research has focused on nanoporous gold (NPG) owing to its ease of manufacture, chemical stability, and biocompatibility [23]. NPG is a 3D nanostructured material with pore sizes that can be tuned over the range 5 nm to greater than 2 µm [14]. The morphology of NPG is generally characterised using atomic force microscope (AFM) and scanning or transmission electron microscopy (SEM/TEM). The electrochemically addressable accessible surface area is evaluated by measuring the roughness factor calculated from the charge associated with reduction of gold oxide in 0.5 M H_2_SO_4_ solution and applying a conversion factor of 390 µC/cm^2^ [24]. NPG electrodes possess good electrical conductivity, catalytic activity, high surface-to-volume ratio, permeability, chemical, and thermal and mechanical stability [25,26], as well as properties of interest for a range of applications including biocatalysis [27], nucleic acid sensors [28], enzymatic sensors [29], non-enzymatic sensors [30], immunosensors [31], supercapacitors [32], enzymatic fuel cells [33], and so on.

Different methods have been studied for the controlled manufacture of nanoporous gold [33]. For example, using anodization methods, the 3D structure is generally formed by the anodization of gold in oxalic acid at different applied potentials, which enables the formation of specific nanoporous structures [34]. Recently, a NPG microelectrode was fabricated via electrochemical anodization-reduction steps in 0.5 M H_2_SO_4_, exhibiting pore sizes in the range of 30–50 nm [35]. Although anodization of gold avoids the use of corrosive chemicals, the pore diameters are typically ca. 20 nm in size [36], making it potentially difficult to achieve high loading of biomolecules. Another route entails using hydrogen bubbles formed via the electrochemical reduction of H^+^ as the template [37]. Gabriella Sanzo et al. synthesized a gold nanocorals porous structure with an electroactive area 500 times higher than a gold screen printed electrode that was used as the base substrate [38]. The nanocoral electrode was modified with glucose oxidase for the development of an enzymatic biosensor based on the detection of H_2_O_2_. The nanocoral electrode showed a sensitivity of 48.3 µA/mMcm^2^, two times higher than that of the bare gold electrode. The hydrogen template produces materials with pore sizes in the micrometer region. In order to overcome the limitation on pore size, other template routes can be used. The hard template route usually involves two steps: assembly of monodisperse spheres, then electrodeposition of the metal followed by removal of the hard template, where the diameter and thickness of the porous structure are controlled in the range of 100–2000 nm [39]. The spatial arrangement and size of the pores can be controlled using colloidal crystals as a template. For instance, Szamocki et al. fabricated macroporous gold electrodes of different sizes for the electrochemical oxidation of glucose with glucose dehydrogenase (GDH), with an enhanced electrochemical response by more than one order of magnitude compared with planar gold electrodes [39]. Gamero et al. immobilised lactate oxidase (LOx) on NPG with a pore size of 500 nm, with a linear response observed up to a concentration of 1.3 mM [40].

An alternative approach relies on chemical dealloying of the less noble metal of an alloy, which can be prepared by sputtering a gold-metal alloy onto a support or by using commercially available gold alloys, for example “white gold”. During the dealloying process, atoms of the less noble metal are detached from the surface and subsequently dissolved under the etchant conditions, forming nanoporous structures. Different alloy systems including Au-Zn [41], Au-Ni [42], Au-Al [43], Au-Si [44], and Au-Ag [45], have been used for the formation of nanoporous gold by dealloying the least noble metal component. Au-Ag is the most commonly used owing to the ease of removal of silver, which is generally removed under corrosive conditions (usually 70% nitric acid). In a systematic study, different alloy compositions Ag_70_Au_30_, Ag_50_Au_50_, and Ag_35_Au_65_ were prepared (Figure 1 A–D) [46]. The silver content in the alloy Ag35/Au65 was too low to enable nanoporous structures to be formed. A homogeneous distribution of nanopores was formed using the Ag_70_Au_30_ alloy. The thickness and composition of the layer were controlled by the sputtering conditions, while the pore sizes were controlled by factors such as the time period and the temperature of the process. For instance, by varying the temperature and time of dealloying of a 100 nm thick Ag_70_Au_30_ alloy, the pore size of the dealloyed sheets ranged from 4 to 78 nm, with a maximum surface area 44 times greater than the geometric area [46]. NPG prepared using this approach exhibits a controllable pore size range from 5 to 700 nm [47], a range sufficiently large to accommodate biomolecules. As with planar gold electrodes, the surfaces of NPG can subsequently be modified. For example, carboxylic acid terminated diazonium salts were covalently attached onto NPG and the immobilization of fructose dehydrogenase (FDH) was subsequently accomplished via crosslinking with CMC [48]. The sensor showed a linear range of 0.05–0.3 mM, with a sensitivity of 3.7 μA/cm^2^ mM and a limit of detection (LOD) of 1.2 μM with a fast response of less than 5 s. The linear range encompasses that observed in juices and the sensor displayed excellent selectivity.

Wearable sensors have the potential to play a major role in the development of continuous monitoring for glucose and other biomarkers in different fluids such as tears, saliva, interstitial fluids, and sweat. Flexible NPG was prepared using an electrochemical dealloying approach. NPG electrodes were further modified with lactate oxidase and bilirubin oxidase to develop a lactate/O_2_ enzymatic fuel cell, which was successfully tested in artificial lachrymal fluids [33].

Matharu et al. described the fabrication of NPG with different pore sizes via dealloying of a 600 nm thick Au-Ag alloy to investigate DNA hybridization in the presence of biofouling species [49]. The thiolated capture probe DNA and its target DNA were used to investigate hybridization using methylene blue as intercalator. In the absence of biofouling conditions, the accessibility of target DNA increased with larger pores, resulting in higher signal suppression with maximum values of ∼70% of that for a pore size of about 45 nm. However, in the presence of biofouling conditions, electrodes with average pore sizes of 25–30 nm maximized the accessibility of target DNA as the pores were sufficiently small to block the entrance of biofouling molecules. In contrast, larger pores were susceptible to electrode blockage by biofouling, decreasing the biosensor performance. 

Owing to the expensive nature of gold, electrodes have been manufactured using lower cost non-noble metals such as Cu, Ni, Ti, or Fe [23]. However, the reproducible preparation of nanoporous structures from such alloys needs to be addressed [50]].

NPG is the most widely used metal support used for biosensing, with reviews on the preparation and application of nanoporous gold published recently [14,51]. The high surface area per volume, biocompatibility, and the ability to prepare flexible electrodes make NPG an attractive material for use with biological systems. However, the high cost of gold and the complexity of the manufacturing process currently limit the applicability of NPG to research applications [43]. 

### 2.2. Carbon Based Materials

#### 2.2.1. Graphene

Graphene is a flat sheet of two dimensional layer sp^2^ bonded carbon that is one atom in thickness [52]. The carbon atoms are sp^2^ hybridized, with out of plane π bonds that are responsible for the high electrical conductivity of the graphene. Graphene is of interest owing to its high specific surface area; electrical conductivity; and thermal, optical, and mechanical properties [53]. These remarkable properties have potential applicability in electrochemical biosensors [54]. In comparison with more traditional carbon materials, graphene has a large theoretical surface area (2630 m^2^/g) [55], higher electrical conductivity (200 S/m) [56], and good mechanical strength (1.0 TPA) [57].

Graphene was first prepared via mechanical exfoliation of highly oriented pyrolytic graphite [52]. Other methods such as the exfoliation and cleavage of graphite, chemical vapor deposition, plasma enhanced chemical vapor deposition, solution based reduction of graphene oxide (GO), and so on have been reported [54]. Each of these strategies produces graphene material with different characteristics. These methods focus on the production of large areas of single layers of graphene at low cost and high scale. The primary obstacle to achieving single or a small number of graphene layers is to overcome the strong, interlayer, van der Waal’s forces. To date, the most common approach to graphite exfoliation is the use of strong oxidizing agents that yield GO, a non-conductive hydrophilic carbon material, in a process known as Hummers method [58]. GO produced via this route can be reduced or used for the immobilization of biomolecules. GO can be also obtained using an improved version of Hummers’ method, with a material that contains fewer defects in the basal plane [59]. Liu et al. reported a glucose biosensor obtained via covalent immobilization between the carboxyl and amine groups of GO and GOx, respectively [60]. A nanocomposite film based on chitosan-ferrocene GO (positively charged) was used to immobilise negatively-charged GOx [61]. The biosensor showed a linear response to glucose in the concentration rage of 0.02 to 6.78 mM, with a sensitivity of 10 µA/mMcm^2^ and an LOD of 7.6 µM. Using thermal, electrochemical, or chemical reduction processes to eliminate oxygen-functional groups (ketone, epoxy, carboxyl, and so on) results in graphene with properties that include excellent electrical conductivity, large surface area, and ease of functionalization. Furthermore, residual functional oxygen groups are available for the immobilization of biomolecules [62].

However, owing to the lack of oxygen functional groups to anchor biomolecules, it is necessary to functionalise graphene. Fenzt et al. anchored 1-pyrenebutyric acid onto graphene and an aptamer against the coagulation factor thrombin was subsequently covalently attached [63]. The biosensor displayed a limit of detection of 1 and 5 pM in buffer and serum, respectively. Lee et al. developed a patch-based strip-type disposable sweat glucose sensor and microneedle-based point-of-care therapy [64]. In addition to the detection of glucose, the wearable device consisted of stretchable sensors for humidity, pH, and temperature. A mixture of graphene, GOx, and chitosan was drop cast onto a gold working electrode, followed by Nafion® (Chemours, US) and subsequently glutaraldehyde to cross-link the enzyme layer. The patch was reusable and could be reattached several times. The response for the detection of glucose was corrected via simultaneous measurement of pH. When tested on human subjects, pre- and post-prandial glucose levels correlated with those obtained using a commercial glucose kit. Multiplexed biosensors aim to detect several target biomarkers by integrating a series of sensors on a chip [65]. Such systems are of assistance for the correct diagnosis/treatment of specific diseases. For instance, it was recently shown that lactate is the most important carrier for cancer cells and diabetic patients are prone to accumulate lactate in their tissue [66], and thus a multiplexed biosensor that can be used to discriminate between diseases would be very beneficial.

Owing to the lack of functional groups to anchor biomolecules, pristine graphene has not been extensively used as a biosensor. Functional doping of graphene with heteroatoms such as N, S, B, P, and F is an excellent pathway to enhance electron transfer processes [63]. Among them, nitrogen-doped graphene (NG) offers better electrochemical activity owing to the positive charge density in carbon atoms adjacent to N dopants, enhancing the conductivity of the material [67]. A multilayer biosensor containing GOx, nitrogen-doped graphene, chitosan, and poly(styrene sulfonate) was constructed layer-by-layer [68]. The presence of NG decreased the charge transfer resistance of the assembly, increased the interfacial capacitance, and provided a film matrix with significant charge separation. The biosensor operated at a low potential of −0.2 V versus Ag/AgCl and exhibited a short linear range between 0.2 and 1.8 mM. Nevertheless, the selective doping of N in specific sites is still a challenge and further research needs to be performed for the development of more reproducible methods of preparation. Reviews on the synthesis, characterization, and applications of NG have been published recently [69,70].

To avoid the loss of electrochemical active area and irreversible π–π stacking aggregation, graphene is generally combined with different nanomaterials (e.g., gold nanoparticles, polyaniline, carbon nanotubes, chitosan, Nafion, methylene green, and so on) to enhance the sensitivity of detection [71]. Recently, a graphene thionine gold nanoparticles (AuNP) composite material was used as a paper-based electrochemical immunosensor for the detection of the cancer antigen 125, a biomarker related to ovarian cancer [72]. An impedimetric HIV-1 biosensor based on graphene-Nafion composite was reported. The decrease in electron transfer resistance was proportional to the concentration of HIV-1 gene over the concentration range 1.0·× 10^−13^ to 1.0·× 10^−10^ M and displayed a limit of detection of 2.3·× 10^−14^ M [73]. A third-generation glucose biosensor was fabricated using a graphene/polyethyleneimine/gold nanoparticle for the immobilization of GOx using glutaraldehyde as a crosslinker. The biosensor displayer a linear response to the concentration of glucose over the range of 1–100 µM with a sensitivity of 93 µA/mMcm^2^ [74]. An enzymatic amperometric sensor based on a graphene/PANI/AuNPs modified glassy carbon electrode was reported [75]. The adsorption of GOx facilitated direct electron transfer between the modified electrode and enzymes. Although adsorbed enzyme molecules retained their activity, the leakage of enzymes is a major drawback, a drawback that can be overcome by encapsulation, otherwise covalent binding of the enzymes may be required. Conductive polymers such as polyaniline, polythiophene, polyacetylene, and polypyrrole have been extensively used for the entrapment of biomolecules. The thickness of the polymer film, and thus the barrier to diffusion, could be controlled by tuning the deposition parameters [76]. Such polymer provides high conductivity, biocompatibility, and high stability. For example, a glucose biosensor based on GOx immobilized onto 3,4-ethylenedioxythiophene microspheres modified with platinum nanoparticles retained 97% of its sensitivity after 12 days of storage at room temperature [77]. Owing to its high surface area, high conductivity, and ease of functionalization, graphene has been used extensively as a platform for the construction of a wide range of biosensors references [78], and it holds promise in the development of biosensors for minimally invasive continuous monitoring in, for example, interstitial fluids. The main source of graphene is graphite, which is inexpensive and readily available. However, issues with the degree of biocompatibility of graphene have yet to be fully resolved. Additional challenges include the development of robust biosensors that can function in a range of operating conditions and the preparation of mechanically robust single-sheet graphene electrodes.

#### 2.2.2. Carbon Nanotubes

Carbon nanotubes (CNTs) are one dimensional (1D) carbon tubes prepared by rolling a graphite sheet of variable length and diameter. CNTs are light and possess a large surface area, excellent conductivity, and good mechanical strength, together with chemical and thermal stability. Thanks to these properties, CNTs can be used as transducer or nanocarrier in biosensors [79]. It has a theoretical surface area of 1315 m^2^/g, 50% of that of a single graphene sheet [80]. CNTs can be divided into two main groups: single-walled nanotubes (SWNTs) and multi-walled nanotubes (MWNTs). SWNT is a single layer nanomaterial formed by rolling a graphene sheet into a seamless molecular cylinder with diameter and length ranging between 0.75–3 nm and 1–50 µm, respectively. MWCNT is composed of at least two layers of graphite sheets, separated by approximately 0.42 nm, with a diameter ranging from 2 to more than 100 nm [81].

Different routes have been developed for the manufacture of CNTs. The main method is chemical vapor deposition (CVD), which is based on the decomposition of a carbon source gas at 600–1000 °C producing CNTs. CNTs can be grown directly on the substrate at large scale and low cost. In spite of the simplicity of the process, the use of a metal catalyst such as Co, Fe, Cu, Cr, and so on is required, which can subsequently be incorporated into defects [82]. Another approach that uses metals as catalysts is based on laser ablation, as reported by Smalley and co-workers in 1995 [83]. Carbon atoms from graphite and metal catalysts atoms are irradiated using high energy laser beam. This method results in high purity materials with few defects, but is expensive with high levels of energy consumption and is not practical for large-scale production. Another high-cost approach is the arc discharge method [84], where CNTs are deposited onto a graphite cathode under the action of a current in a vacuum reactor. 

Although the structural integrity of enzymes is preserved via non-covalent functionalization, the interaction between enzymes and CNTs is weak, resulting in leakage of the enzyme. This limitation can be overcome by functionalizing the CNT surface or using nanoparticles or polymers for enzyme immobilization [85]. Paolo et al. electrochemically grafted 2-aminoantrhracene diazonium salt onto SWNCT-based electrodes that were further incubated in a solution of FDH [86]. The biosensor displayed a linear range from 0.05 to 5 mM, a sensitivity of 47 µA/mMcm^2^, an LOD of 0.9 µM, and great stability (90% of retained signal after 60 days). A Pt electrode was modified with a rGO/CNT/AuNPs composite for the detection of lactate. At a potential of 0.2 V, the sensor had a wide linear range of 0.05 to 100 mM with high sensitivity (35.3 µA/mMcm^2^) and a low LOD (2.3 µM) [87]. A wearable glucose biosensor was prepared by immobilizing GOx onto SWCNTs with Nafion® (Chemours, US), which could detect glucose with a response time of less than 5 s [88]. The response to glucose was transmitted to a smartphone using a wireless connection and a linear response to glucose over the range 0.05 to 1 mM was observed.

However, it is important to remark that toxic effects, mainly owing to the presence of metallic impurities, can occur with CNTs. Further studies require the creation of biocompatible CNT-based electrodes that can be addressed by adding dialysis bags [89] or by coating with biocompatible polymers (e.g., chitosan, collagen, Nafion® (Chemours, US), and so on) [90]. CNTs have been also successfully tested for a wide range of biomolecules such as DNA [91], immunosensors [92], proteins [93], and other biological molecules. Graphene and CNTs possess high thermal, mechanical, and electronic properties and both materials can be produced on a large scale. However, the synthesis of CNTs is a high cost process and usually involves the use of metal nanoparticles, which can be toxic, limiting its potential use. 

#### 2.2.3. Ordered Mesoporous Carbon

Ordered mesoporous carbon (OMC) is a flexible material that provides interconnected channels for the diffusion of electroactive compounds in electrochemical systems. OMC possess high specific surface area; large and open porous structure; high conductivity; and excellent chemical, thermal, and mechanical stability [94]. OMC can be synthesized via catalytic activation of carbon precursors, carbonization of the blends of one thermosetting precursors and one thermally unstable polymer, and carbonization of organic aerogels. Nevertheless, the resulting mesopores have abundant micropores and a wide pore distribution [95].

A more reliable pore size distribution with a symmetric ordering can be obtained through a template method, which can be subdivided into two categories: hard and soft-templating. In hard-templating, the pore size is controlled using a mesoporous silica template that controls the pore size. Other templates such as nickel oxide can also be used. The overall process involves the impregnation of the pores of the template with a carbon source (e.g., sucrose, ethylene, furfuryl alcohol), followed by polymerization and carbonization upon pyrolysis. Dissolution of the silica in HF or alkaline solution results in a mesoporous carbon replica. The largest used mesoporous silica are the hexagonal SBA-15 and cubic MCM-48 materials, leading to CMK-3 and CMK-1 respectively [94]. The structure of CMK-1 is dependent on the carbon precursor used. CMK-3 exhibited a highly ordered hexagonal packed mesopores interconnected owing to the presence of micropores. The process for the formation of OMC via hard-templating is schematically represented in Figure 2A. The pore arrangement of CMK-1 carbon replicas resulted in a more accessible structure owing to the more favorable rate of diffusion of reactant molecules during catalytic processes [96]. CKM-1 modified with an ionic liquid showed a good electrocatalytic response for the direct oxidation of dsDNA with a detection limit of 1.2 µg/mL [97]. A CMK-3 was used for the construction of alcohol and glucose biosensors, based on alcohol dehydrogenase and glucose oxidase [98]. To date, the majority of OMC biosensors rely on the use of CMK-1 and CMK-3 [94]. Other mesoporous ordered silica have been used as a hard template for the synthesis of OMC. For instance, a 1D-carbon nanotube array, designated as CMK-5, was synthesized when the channels of SBA-15 were partially filled [99]. The covalent immobilization of GOx was performed using a 4-nitrophenyl functionalized CMK-5, exhibiting a linear response over the range of 1–14 mM [100]. The electrochemical response of the sensor was reduced by 6% after one month of storage.

The hard-template route requires the use of expensive reagents for the impregnation of the template and toxic reagents such as HF for the selective removal of the silica template. Besides, it is a time-consuming and multi-step complex process and, consequently, the manufacture of OMC at high scale is not suitable [101]. Efforts have been made for the development of OMC at cost-effective approaches with controllable pore size. Via soft-templating, OMC is obtained via self-assembling of supramolecular aggregates of carbon precursors (thermosetting agents such as phenolic resin or resorcinol-formaldehyde mixtures) and an amphiphilic copolymer surfactant (F127, CTAB, P123, and so on) as a template (Figure 2B) [102]. The carbon precursor was polymerized to form a highly cross-linked composite, followed by the template removal and carbonization. The direct process is simple, cost-effective, and suitable for large-scale industrial applications [103]. Using F127 as a soft-template, a GOx-based biosensor displayed a linear concentration range from 5 to 100 mg/mL [104].

More recent developments are focusing on the preparation of OMC with large mesopores and graphite walls. A hierarchically porous partially graphitic carbon membrane with three-dimensionally networked nanotunnels was used as a monolithic electrode matrix for the construction of a glucose biosensor [105]. The nanotunnels (~40–80 nm in diameter) are composed of partially graphitic carbon with ordered mesoporous (~6.5 nm in diameter). The carbon material was subsequently modified with polydopamine and decorated with AuNPs for the immobilization of GOx. The biosensor displays an LOD of 4.8 pM, which is four orders of magnitude lower than conventional nanostructured enzymatic glucose sensors. 

High surface volume and ordered mesoporous make OMC an interesting material for biosensing, although they still suffer from a number of limitations. The removal of silica or polymer requires the use of HF, NaOH, or high temperature [106]. OMC materials are usually powdered materials and the use of a binder is required in order to improve the mechanical stability that can be tackle with the development of continuous OMC directly attached onto the electrode surface. Finally the majority of studies to date rely on the use of CMK-3 or CMK-1, materials that are not suitable for production on a large scale. Future studies require the development of alternative OMC materials that could provide the same advantages of graphene or carbon nanotubes.

## 3. 3D-Printing Technology

Additive manufacturing (AM) or three-dimensional printing is an emerging eco-friendly technology that holds promise to revolutionize the fabrication process. AM is based on layer-by-layer deposition of materials onto a substrate capable to manufacture geometrically complex objects in a one-step digitally controlled process [107]. 3D-printed devices are manufactured in a highly flexible manner with fast process times, generating minimum waste while offering precise replication and reducing constraints of creativity. In contrast, conventional technologies require complex, expensive machinery and tools (drilling, milling, and so on) [108]. The specific applications and requirements (material, composition, transparency, and so on) of the printed device define the most suitable 3D-printer technology. To date, various 3D-printing processes have been examined, including fused deposition modeling (FDM), inkjet and polyjet printing, and selective laser melting (SLM) [109], for the development of biosensors. A summary of the processes, printable materials, build volume, advantages, limitations, and printers, as well as cost and printing effectiveness, has been recently described [110,111]. The main applications of 3D-printing technology in the development of electrochemical biosensors are based on the development of fluidic platforms, electroactive, and catalytic surfaces and the manufacture of structures that include 3D-electrodes, flow channels, and auxiliary structures such as microneedles [112].

In comparison with traditional techniques for the manufacture of thin or thick electrodes (focused ion beam milling, electron-beam lithography, photolithography, and screen-printing), AM holds promise in overcoming issues such as high equipment and process costs. Screen-printing is a commonly used approach in the preparation of electrodes. In contrast to screen-printing, AM minimizes the consumption of materials to be printed, and thus reduces waste. AM also allows for the formation of small sized electrodes and the deposition of biomolecules with high spatial resolution. To date, the majority of 3D-printing methods use stainless steel owing to its cost-effectiveness and its passivated surface. In order to make 3D-printed stainless steel suitable for electrochemical sensing, the steel needs to be coated with another metal (Au, Bi, IrO2, Pt, and so on) [109]. For instance, Ambrosini et al. used selective laser melting for the manufacture of 3D-printed stainless steel electrodes, which were then modified with three different catalysts via electrodeposition [113]. A similar approach was also reported by Pumera and co-workers on the use of 3D-printed helical-shaped stainless steel electrodes that were subsequently electroplated with gold [114]. Gold-plated 3D-printed electrodes were utilized as a platform for DNA hybridization with different target DNA sequences (Figure 3). Upon hybridization with complementary DNA, the biosensor displayed a linear response over the range of 1–1000 nM. The selectivity of the sensor was examined using a non-complementary DNA sequence, resulting in a similar electrochemical response to the probe DNA owing to ineffective levels of hybridization. DNA biosensors require the selective discrimination of single-nucleotide polymorphism [115], and thus further investigations of 3D-printed biosensors are needed.

3D-printed metal electrodes are expensive and offer a limited electrochemical potential window that can restrict applications in biosensing. Carbon-based materials are more attractive materials owing to reduced costs. Carbon nanotubes, graphene, and carbon black are commonly used for the development of 3D-printed electrodes. However, 3D-printed carbon electrodes suffer from poor electrochemical performance as the carbon material is combined with polymeric binders, often in the presence of surfactants [116]. The presence of high levels of binder leads to printing issues owing to the high viscosity and tackiness of the ink, whereas low concentrations of binder may result in film cracking [117]. Different methods have been used to improve the electrochemical performance by removing the protective polymer of the top layer and exposing the carbon materials to solvents such as dimethylformamide or by electrochemical activation. Using both methods can enhance the electrochemical performance [118]. Katseli et al. described a carbon black/PLA electrode modified with GOx and Nafion® (Chemours, US) [119]. The glucose biosensor relied on the detection of H_2_O_2_ and exhibited a linear response over the range of 2–28 mM. A 3D-printed graphene/PLA was modified with AuNPs and horseradish peroxidase for the electrochemical detection of hydrogen peroxide at 0.0 V versus Ag/AgCl via direct electron transfer [120]. The biosensor was used to detect H_2_O_2_ in human serum and had a stable response after 7 days of incubation. A 3D-printed graphene/PLA was treated with DMF before the immobilization of GOx by crosslinking with glutaraldehyde [121]. The biosensor relied on the use of ferrocene-carboxylic acid as mediator for the indirect detection of H_2_O_2_ generated from the enzymatic reaction and was utilised for the detection of nitrite and uric acid in human saliva and urine respectively. A review on 3D-printed electrochemical sensors has recently been published [122].

## 4. Conclusions and Future Perspective

Electrochemical biosensors can be readily incorporated into miniaturized, portable devices. Although biorecognition elements provide reliability and good analytical performance, they can suffer from disadvantages such as high cost, short lifetime, and low levels of stability. The preparation of more stable biorecognition elements using a range of genetic engineering approaches to overcome these limitations is a major focus of current research. The development of structured materials with properties tailored to the effective and selective immobilization of the biorecognition elements will be needed for each particular system. The point of care detection of small molecules such as glucose, lactate, cholesterol, and so on has been successfully demonstrated. However, the detection of large molecules such as proteins and nucleic acids suffers from issues such as electrode fouling and non-specific adsorption of biomolecules. Resolution of these issues would enable more widespread use of electrochemical techniques. In the analysis of large biomolecules, low limits of detection are required, levels that can be enhanced with the use of nanomaterials such as nanoporous gold, graphene, or carbon nanotubes. The pore size of nanoporous gold can be tailored in the range 5 to 700 nm, a size range sufficient to accommodate large amounts of biomolecules. Nanoporous gold possesses advantages such as high surface area, good conductivity, and biocompatibility that make it an attractive material for biosensing. However, the complexity of the manufacturing process currently limits the applicability of NPG to research applications. In comparison, the manufacture of carbon nanomaterials can be performed at a relatively low cost. Graphene has enhanced sensitivity for a wide range of biomolecules when compared with other carbon materials such as carbon nanotubes or ordered mesoporous carbon. The unique properties of graphene (high conductivity, high surface area, excellent mechanical properties, ease functionalization, and scalability) and the low cost of manufacture make it an attractive material for the manufacture of biosensors. The use of graphene in non-invasive biosensors can open up new applications in wearable sensors and personalized health. However, such devices face challenges such as improved comfort and analytical performance.

Additive manufacturing has the ability to produce geometrically complex devices in a digitally controlled process. Additive manufacturing methods have been used to prepare a range of structures and electrodes. However, the exploitation of additive manufacturing is still at an early stage and further, detailed investigations are required. For example, the immobilization of enzymes deep within pores and channels may give rise to issues with substrate transport. The use of 3D-printed customized microfluidic devices can potentially overcome such transport limitations. To date, the electrochemical performance of 3D-printed electrodes shows diminished performance when compared with electrodes manufactured using more established methods. The high cost of consumables and instrumentation needed to prepare 3D-printed metal electrodes, and the difficulty in manufacturing porous structures, will possibly limit applications in biosensing. 3D-printed carbon electrodes hold more promise owing to their low cost, ease of fabrication, and suitability for large-scale production. To date, only a relatively small number of biosensors based on 3D-printed electrodes has been reported. Further research is required to produce 3D-printed electrodes at a large scale and with the performance required for clinically relevant analytes.

## Figures and Tables

**Figure 1 sensors-20-03561-f001:**
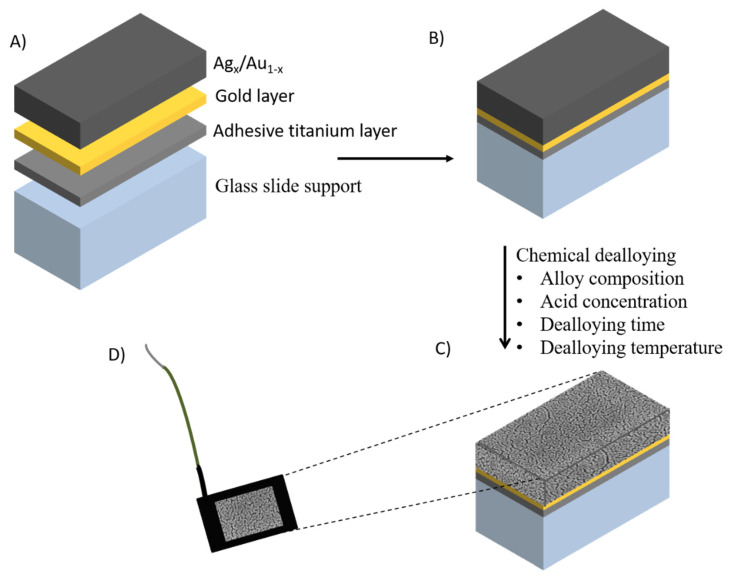
Schematic representation of the manufacture of nanoporous gold (NPG) electrodes with (**A**) different layers and thicknesses, (**B**) sputtered glass sheet prior to etching, (**C**) formation of nanopores after etching, and (**D**) the completed NPG electrode. Adapted from [46].

**Figure 2 sensors-20-03561-f002:**
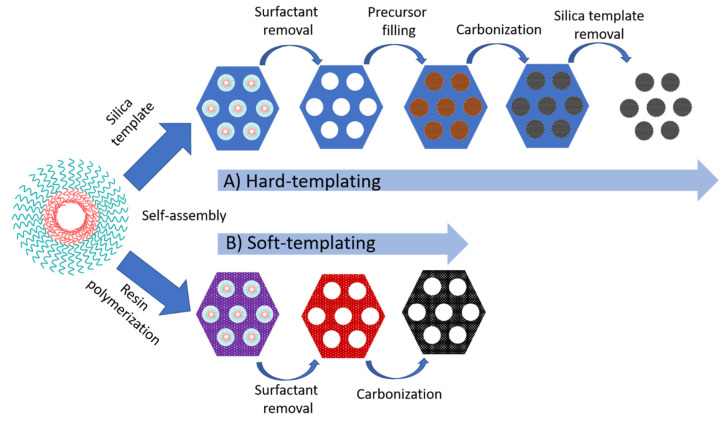
Two typical methods for the preparation of ordered mesoporous carbon materials: (**A**) the nanocasting strategy from mesoporous silica hard templates and (**B**) the direct synthesis from block copolymer soft templates. Adapted from [95].

**Figure 3 sensors-20-03561-f003:**
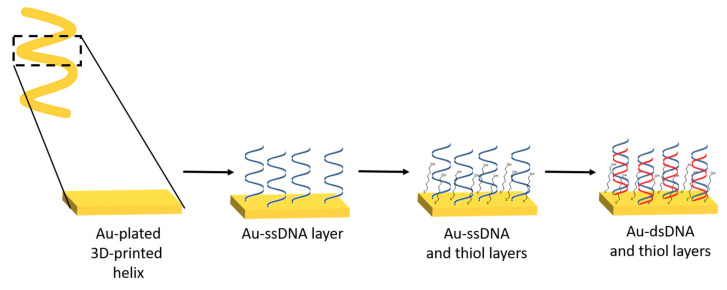
Schematic representation of the preparation of DNA biosensor. The thiolated DNA was covalently immobilized onto a gold-plated 3D-printed helix electrode. The modified electrode was then incubated with a DNA target, and the electrode was then exposed to methylene blue. Adapted from [114].

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
