# Peer review of "Biosensors—Recent Advances and Future Challenges in Electrode Materials"

_sensors, 2020, doi:10.3390/s20123561_

Round 1
Reviewer 1 Report
This review aims at discussing the challenges in electrode materials for application as electrochemical biosensors, focusing primarily on porous or nanoscale materials (nanoporous metals and carbon, graphene and carbon nanotubes, or materials obtained by 3D printing). The manuscript is well written and of potential interest to the readers of Sensors. However, many reviews are already available on the use of nanomaterials for these purposes, some of them being cited here and some not, and the authors should put their work in this context (what is newly described here? This a main concern that should be addressed). Some key papers are missing (e.g., for porous metals prepared by dealloying: J. Mater. Chem. 22 (2012) 11950).
Author Response
We appreciate the positive comments from the reviewer. We emphasise that the focus of the review is on porous nanomaterials, not on nanomaterials per se. As suggested by the reviewer, we have included two new references (J. Mater. Chem. 22 (2012) 11950) and Adv. Eng. Mater. 2017, 19, 1700389) that now correspond to references 21 and 22 (page 2 line 88).
Reviewer 2 Report
Please see in attached file my comments.

Author Response
We thank the reviewer for his/her comments and suggestions. As requested by the reviewer, we have made the following changes:
- The references were modified:
“Ref 10 is Pharmacol. Res.”
“Ref 14 is Bioelectrochemistry”
“Ref 19 is Anal. Biochem.”
“Ref 36 is Bioelectrochemistry” (now referenced as 38)
“Ref 64 is Bioelectrochemistry” (now referenced as 68)
“Ref 67 is Bioelectrochemistry” (now referenced as 71)
“Ref 75 is Chem. Phys. Lett.” (now referenced as 83)
“Ref 88 is Micropor. Mesopor. Mater.” (now referenced as 96)
“Ref 103 is Appl. Mater. Today” (now referenced as 111)
“Ref 110 please remove the “chemistry” “(now referenced as 118)
References 21, 22, 65, 66, 73, 74, 76 and 77have been also added
- A section on multitasking biosensors was added, page 5 lines 216-220 “ (see references 65,66)
“Multiplexed biosensors aim to detect several target biomarkers by integrating a series of sensors on a chip [65]. Such systems are of assistance for the correct diagnosis/treatment of specific diseases. For instance, it was recently shown that lactate is the most important energy carrier for cancer cells and diabetic patients are prone to accumulate lactate in their tissue [66] and thus, a multiplexed biosensor that can be used to discriminate between disease types would be very beneficial”
- Examples of nanomaterials for the modification of graphene were cited in page 6 lines 235-236 “(e.g. gold nanoparticles, polyaniline, carbon nanotubes, chitosan, Nafion, methylene green, etc.)”. More examples of biosensors based on modified graphene were reported in page 6 lines 239-245 “An impedimetric HIV-1 biosensor based on graphene-Nafion composite was reported. The decrease in electron transfer resistance was proportional to the concentration of HIV-1 gene over the concentration range 1.0·10-13 to 1.0·10-10 M and displayed a limit of detection of 2.3·10-14 M [73]. A third-generation glucose biosensor was fabricated using a graphene/polyethyleneimine/gold nanoparticle for the immobilization of GOx using glutaraldehyde as a crosslinker. The biosensor displayed a linear response to the concentration of glucose over the range 1 – 100 µM with a sensitivity of 93 µA/mMcm2 [74]. “ (see references 73 and 74).
- A section on conductive polymers was added, page 6 lines 250-256
“Conductive polymers such as polyaniline, polythiophene, polyacetylene and polypyrrole have been extensively used for the entrapment of biomolecules. The thickness of the polymer film and thus the barrier to diffusion, could be controlled by tuning the deposition parameters. Such polymers provide high conductivity, biocompatibility and high stability. For example, a glucose biosensor based on GOx immobilized onto 3, 4-ethylenedioxythiophene microspheres modified with platinum nanoparticles retained 97% of its sensitivity after 12 days of storage at room temperature” See references 76, 77
- A coma after polyjet printing, was added (after modifications, this change is located in page 9 line 378)
- As suggested, the conclusions of the section 3 (3D-printing technology) was removed
The conclusions were altered (page 10 lines 445-447)
“Although biorecognition elements provide reliability and good analytical performance, they can suffer from disadvantages such as high cost, short lifetime and low levels of stability. The preparation of more stable biorecognition elements using a range of genetic engineering approaches to overcome these limitations is a major focus of current research. The development of structured materials with properties tailored to the effective and selective immobilisation of the biorecognition elements will be needed for each particular system ”
Reference 8 was changed, as the original reference was incorrect, from “Bioelectrochemical sensing devices have been effectively transferred from the laboratory to the point-of-care (POC) with global sales growing from less than $5 million per annum thirty years ago to over $18 billion in 2018 [8]” to “Bioelectrochemical sensing devices have been effectively transferred from the laboratory to the point-of-care (POC) with global sales growing from less than $5 million per annum [8] thirty years ago to over $18 billion in 2018”.